# How do children overcome their pragmatic performance problems in the true belief task? The role of advanced pragmatics and higher-order theory of mind

Lydia Paulin Schidelko *, Marina Proft, Hannes Rakoczy

Department of Developmental Psychology, University of Göttingen, Göttingen, Germany

* lydiapaulin.schidelko@uni-goettingen.de

**Data Availability Statement:** All relevant data are within the paper and its Supporting Information files.

## Abstract

The true belief (TB) control condition of the classical location-change task asks children to ascribe a veridical belief to an agent to predict her action (analog to the false belief (FB) condition to test Theory of Mind (ToM) abilities). Studies that administered TB tasks to a broad age range of children yielded surprising findings of a U-shaped performance curve in this seemingly trivial task. Children before age four perform competently in the TB condition. Children who begin to solve the FB condition at age four, however, fail the TB condition and only from around age 10, children succeed again. New evidence suggests that the decline in performance around age four reflects pragmatic confusions caused by the triviality of the task rather than real competence deficits in ToM. Based on these results, it can be hypothesized that the recovery of performance at the end of the U-shaped curve reflects underlying developments in children's growing pragmatic awareness. The aim of the current set of studies, therefore, was to test whether the developmental change at the end of the U-shaped performance curve can be explained by changes in children's pragmatic understanding and by more general underlying developmental changes in recursive ToM or recursive thinking in general. Results from Study 1 ($N = 81$, 6–10 years) suggest that children's recursive ToM, but not their advanced pragmatic understanding or general recursive thinking abilities predict their TB performance. However, this relationship could not be replicated in Study 2 ($N = 87$, 6–10 years) and Study 3 ($N = 64$, 6–10 years) in which neither recursive ToM nor advanced pragmatic understanding or recursive thinking explained children's performance in the TB task. The studies therefore remain inconclusive regarding explanations for the end of the U-shaped performance curve. Future research needs to investigate potential pragmatic and general cognitive foundations of this developmental change more thoroughly.

## Introduction

Theory of Mind (ToM) is the social-cognitive ability to think and reason about one's own and others' mental states [1]. At the core of ToM lies meta-representation: the capacity to represent

**Funding:** The reported project received financial support from the Deutsche Forschungsgemeinschaft (DFG, German Research Foundation) by employing the first author on Research Project 254142454 / GRK 2070. The funders had no role in study design, data collection and analysis, decision to publish, or preparation of the manuscript.

**Competing interests:** The authors have declared that no competing interests exist.

that subjects represent the world in a certain way that can differ from one's own current perspective. The litmus test for ToM is the so-called false belief (FB) task. In this task, participants need to track a story protagonist's belief that comes to differ from reality: Participants see how an object is placed at location 1 in the presence of a protagonist and is then moved from location 1 to a new location 2 in the protagonist's absence. Participants are then asked to ascribe to the protagonist a belief about the object's location ("Where does she think the object is?") or to predict the protagonist's behavior based on her belief ("Where will she look for her object upon her return?) [2]. Individuals with an (explicit) ToM predict that the protagonist will look for her object–based on her false belief–in location 1. Developmentally, children succeed FB tasks around the age of four years. Children younger than four years systematically fail. They predict confidently that the protagonist will look–according to reality–in location 2 [2, 3].

In the history of ToM research, many FB studies also administered an additional True Belief (TB) condition to the children. The TB condition serves as a baseline and control measure to ensure that children, especially younger ones, can cope with the narrative task structure. It is structurally similar to the FB task with the only difference that the protagonist witnesses the object transfer from location 1 to location 2 and thus has a veridical belief about the object's location. Typically, children younger than four years who fail the FB condition, pass the TB condition [3].

Recently, however, the TB condition has been administered to a broader age range of children, with quite puzzling results: with age, children get *worse* in the TB condition. More specifically, children from age four who begin to solve the FB task start to fail the TB task [4–7]. The initial pattern found in younger children–passing the TB condition while failing the FB condition–reverses around this age. Children from age four succeed in the FB condition but fail the TB condition. This performance pattern (younger children pass TB and fail FB and vice versa for older children), reveals itself also at an individual level, in strong negative correlations of the two versions of the tasks. Remarkably, the failure in the TB task persists into late childhood: Only from age seven to ten, performance in the TB task recovers. At this point in development, children pass both FB and TB for the first time. Taken as a whole, the development of performance in the TB task follows a U-shaped trajectory: from high performance in young children around age three to a dramatic decrease around age four when children solve the FB condition to a recovery of performance only in later childhood around age seven to ten. FB performance, in sharp contrast, remains constantly high from around age four [6].

As any U-shaped curve in performance, this unexpected developmental pattern raises at least two fundamental questions: First, how does the decrease in performance in the TB task come about at the beginning of the U-shaped curve? Why do children start to fail TB tasks once they come to master FB tasks? Second, how does the recovery of performance come about at the end of the U-shaped curve? Why do children from around age seven to ten overcome their intermediate difficulty with TB tasks?

One possible answer to both questions is the following: The developmental pattern of the U-shaped performance curve in the TB task does not reflect children's ToM competencies but the development of an understanding of pragmatics. Children's failure in this intermediate state between four and ten is not based on a fundamental problem in ToM understanding but on pragmatic performance limitations [8].

In general, pragmatics pertains to comprehension and production of speech acts and discourse that goes beyond mere literal meaning. For a comprehensive understanding of most speech acts and discourse, additional information besides the literal meanings of the words (sentence meaning) needs to be taken into account in order to determine what is meant (speaker meaning): for example, information about who made an utterance, in which context, against the background of which rules etc.–and in particular, the recipient needs to figure out

the speaker's intentions underlying the speech act in question [e.g., 9]. Pragmatics is thus, in some sense, a form of applied ToM.

Regarding the TB performance, it seems quite clear that children from early on do understand the semantics, the literal meaning of the words in the TB test question. Perhaps, however, children in the intermediate state (between ages four and ten) struggle with understanding the use of the test question "Where does the protagonist think the object is?" They do not yet understand what the interlocutor means or wants by asking this question. But why should the TB question be pragmatically challenging? Why do children struggle to grasp the experimenter's intention in asking the TB test question?

A closer inspection of the TB task and the corresponding question reveals that it combines a number of properties that jointly make the task quite peculiar from a pragmatic point of view. First, the TB task is highly trivial: The protagonists clearly sees that the object is moved to a new location and the protagonist, the child and the experimenter share this knowledge about where the object is and everyone knows that the others know, too (this is thus common ground or mutual knowledge). Second, the TB test question is an academic question. The experimenter knows the answer herself. She does not ask this question to gain new information but rather to test whether the child knows the answer, too. Academic question formats are difficult to grasp for young children [10] (for effects of the interviewer's knowledge on children's answer behavior, *see also* [11–13], for related proposals regarding the role of pragmatic factors in FB and other ToM tasks, see, [14–17]. Third, the TB test question asks for a belief ascription or a belief-based action prediction [6]. Normally, we tend to talk about beliefs when we refer to or at least raise the possibility of their falsity [18]. In the TB scenario, however, there is no such obvious possibility. From a purely semantic point of view, the TB test question is utterly unproblematic, indeed highly trivial. Children with a merely literal language use with little sense of pragmatics should thus not find such questions taxing. But for language users with some pragmatic sensitivity, such questions should appear at least *prima facie* odd.

In light of the first question raised above (How does the decrease in performance in the TB task come about at the beginning of the U-shaped curve?), the pragmatic analysis thus yields the following (somewhat simplified) picture: Young children without a sophisticated understanding for ToM are limited in their pragmatic language understanding. They mostly use and interpret language literally (but *see* [19]). Children in this stage of development thus should have no problems with the TB task. However, once children start to develop ToM capacities (i.e., when they pass the FB task), these lay the ground for developing pragmatic understanding [20, 21]. However, their initial ToM and pragmatic understanding are still fragile at this age and their fragile pragmatics then leads them astray in the TB test.

Some evidence speaks in favor of this. First, as reviewed above, performances in the TB and FB tasks are negatively correlated such that children first fail FB and pass TB, and then show the reverse pattern [6]. The performance pattern matches the predictions of the pragmatic analysis such that children's failure in the TB task depends on their success in the FB task: once children develop the prerequisite ToM capacities, they develop an understanding for pragmatics. As a consequence of this development, they suffer from the pragmatic peculiarity of the TB question and fail to answer it correctly while passing the FB task [6]. Second, children succeed in both the TB and FB task after modifications of the task pragmatics in the TB task. For example, children solve the TB task without any decline in the performance curve when the task is presented without or with less trivial language [8]. These results suggest that children show no more difficulties with answering the TB test question correctly once peculiar factors of the tasks are removed. Taken together, first evidence confirms the predictions of the pragmatic analysis at the beginning of the U-shaped performance curve.

The developmental processes at the end of the U-shaped performance curve, however, still raise open questions. How do children overcome their pragmatic difficulties in the TB task later in development, and how does this explain performance recovery at the end of the U-shaped curve? Currently, hardly anything is known about how children overcome their performance limitation in the TB task. From a pragmatic point of view, one possibility is the following: Regarding the beginning of the U-shaped development, the pragmatic analysis predicts that children start to get confused by the peculiarity of the TB test question once they develop sensitivity for pragmatics on the basis of their developing ToM. Applying the same logic to the end of the U-shaped curve, the pragmatic analysis predicts that children succeed again in the TB task once they have undergone further pragmatic development (adults, after all, even though they may find this type of questions funny, have no difficulty in answering it correctly). Taking new steps in pragmatics development might enable children to reason about language use on a higher, more sophisticated level of pragmatic interpretation, and thus to grasp more complex and advanced forms of discourse and speech acts.

Imagine, to illustrate the point, someone asks you, "Did you enjoy the nice weather yesterday?" when it in fact was raining cats and dogs all day. Interpreting this as a regular question asked in order to receive new information would be highly confusing given that the presupposition ("The weather was nice yesterday") is not fulfilled. To understand the actual speaker meaning, you need to stand back from the literal meaning of the question and interpret it at a different, higher level. For example, you might infer the speaker's intention to make an ironic comment about the rainy weather, or you may interpret the question as an academic exercise such that the speaker's intention is to test your knowledge of English past tense. What these examples thus illustrate is that a *prima facie* odd question can make perfect sense when interpreted on a new and higher pragmatic level [22].

Applied to the TB performance, this developmental step might enable children to focus on a new interpretational level that allows them to resolve their initial confusions about the peculiar test question in the TB task ("strange question, but then I'm participating in a study after all; researchers do ask strange questions..."). If this idea holds, children who are able to reason at a flexible, higher-order level of pragmatic interpretation should be more likely to answer the TB test question correctly. Accordingly, the emergence of flexible, higher-order pragmatic understanding would determine the end of the U-shaped performance curve.

But what exactly happens in children's pragmatic development at the end of the U-shaped curve? How can the crucial pragmatic development at the end of the U-shaped curve be described, and what are important foundations and correlates of this development?

The pragmatic analysis predicts a developmental progress in advanced pragmatics at the end of the U-shaped curve. If indeed the developmental curve reflects pragmatic progress more generally, this should also become apparent in other areas of advanced pragmatics. To this end, we will compare children's performance in the TB task at the end of the U-shaped curve with developments in *advanced pragmatic understanding* more generally (regarding comprehension of indirect speech acts). The progress in advanced pragmatics, in turn, could itself be rooted in growing *recursive ToM capacities*. And the development in higher-order recursive ToM, in turn, might be based on a more *general ability for recursive thinking*. In the following, we discuss these three possibilities in turn.

## Advanced pragmatic understanding

The crucial step in pragmatic development that leads to the end of the U-shaped curve may reflect a more general phenomenon of developmental progress in pragmatics. Such a general progress in advanced pragmatics might become evident when comparing the performance in

the TB task and advanced pragmatics in other areas. Generally, pragmatics is neither a simple and unitary phenomenon [23] nor do pragmatic abilities emerge in simple uniform ways across ontogenetic development [24]. Most relevant for present purposes are *advanced* forms of pragmatic understanding that tend to emerge comparatively late in development. Non-literal language understanding, such as ironic and metaphorical utterances, are prototypical examples [21, 25, 26]. For example, in uttering "It's great weather for our picnic today" when it is raining all day, the speaker does not want the hearer to take her utterance literally. The speaker rather intends the hearer to belief that she thinks that it is not nice to have a picnic outside [27, p.262]. Accordingly, the hearer needs to suppress the initial, literal interpretation to be able to infer the actual meaning, for example, by taking context information into account [24]. Similar processes may be at work in the TB task such that children might overcome their confusion about the trivial test question ("This is too easy, maybe I missed something") by suppressing this initial interpretation and taking context information into account ("I'm participating in a study and the experimenter asks test questions"). Interpreting non-literal language, especially ironic utterances, involves ascriptions of complex intentions and is therefore suggested to be an application of ToM [20, 26]. In neurotypical development, children develop an understanding of metaphors during school age or even preschool age [19, 24, 28]. Its relation to ToM abilities, however, has been controversially discussed in the recent literature [28–30]. Evidence on irony comprehension suggests that children develop an understanding of ironic utterances during school age between six and 13 years of age, depending on the kinds of measures used [24]. The relation of irony understanding to ToM is less controversial: children's performance in irony comprehension correlates with their second-order FB understanding [26, 31].

## Recursive ToM

The crucial foundation of development regarding the end of the U-shaped curve might be even more general than sophisticated pragmatics–for example, the capacity for recursive, higher-order mindreading. The standard ToM tests asks for a first-order mental state ascription, but of course this is not where mental state ascription ends. Advanced forms of ToM enable flexible and higher-order forms of mindreading in which additional levels of mental states can be represented. Ascription of a belief about a belief, for example, constitutes second-order mental state ascription. Virtually, mental states can be recursively embedded within each other infinitely ("A thinks that B thinks that C thinks that D thinks that . . . p"). This understanding for recursively embedded mental states may be the common denominator underlying both advanced pragmatics generally and of the TB performance specifically. It may provide the basis for advanced pragmatics in various forms, for example, by enabling the ascription of higher-order communicative intentions. In the TB task, recursive ToM might enable the ascription of specific higher-order communicative intentions to make sense of the speech act. There are at least two possibilities, how recursive ToM might be involved in the TB task in particular. One possibility is that children who develop an understanding for the pragmatic use of academic test questions (which are also very trivial and pertain to subjective representations) do not any longer get confused by the TB test question at all, but reach the right pragmatic interpretation of the question straight away. They integrate context information and ascribe higher-order intentions and thus make sense of the academic test question from the outset without being confused. A second possibility is that children initially suffer from confusion by the trivial academic TB test question (e.g., "Why does the experimenter ask me such a question? Maybe I must have missed something, maybe the experimenter thinks that I don't know that the protagonists does not know that the object is in that location"). Younger

children might not yet be able to resolve this confusion whereas older children, once they have developed recursive mindreading, might be able to ascribe higher-order mental states and thus to resolve the confusion (e.g., "She wants to know whether I know that the protagonist knows that the object is in the new location"). Either way, children's performance on the TB test question should then be related to their recursive ToM capacities more generally.

Recursive ToM reveals itself in various forms, in many different tasks and situations, but tasks that directly test for this understanding for second- or higher-order mental state ascriptions are rare in the literature. Evidence from this line of research shows that children at the age of five to six years can attribute second-order mental states ["A thinks that B believes that . . ."; e.g., 32], but only very little is known about children's development of higher-order ToM beyond second order of recursion. In a study by Liddle and Nettle, for example, ten- to eleven-year-olds performed above chance level in a third-order ToM task and at chance level in a fourth-order ToM task [33]. Adults, in contrast, were able to reason until seventh-order of recursion, in particular, when tested "implicitly" through observing video clips of social interactions compared to explicit measures used in the above reported studies with children [34]. Until now, it therefore remains an open question when exactly children learn to reason about mental states on different levels of embedding and whether this development is fundamental for children's performance in the TB task.

## General ability for recursive thinking

The ability to reason about higher-order mental states, in turn, might be based on a more general ability for recursive thinking. The performance in the TB task, hence, would not only rely on the ability for higher-order ToM but on an even broader ability for recursive operations that is fundamental for higher-order ToM abilities.

Thinking and reasoning recursively is not only of importance in the development of ToM but has been implicated in a number of probably uniquely human abilities such as language, music, mathematics or mental time travel [35–37]. Recursive operations in all these areas require embedding of elements (e.g., mental states, words/clauses, etc.) within elements of the same kind [38]. The corresponding level of reasoning can be more or less clearly defined and quantified (e.g., in the domain of ToM: "A thinks that B thinks that C thinks that p" as third-order mental state ascription).

The general ability for recursive operations might manifest itself in recursive thinking in the various specific areas of application, including higher-order ToM. Consequently, the developmental changes in TB performance might reflect development in advanced pragmatic that builds on recursive ToM that, in turn, is a manifestation of general recursive operations. Once the child has acquired a certain level of general recursive thinking, this enables her–via recursive ToM–to think pragmatically about the TB task at a higher level, overcome her pragmatic confusion and solve the task. If this hypothesized pattern holds, an individual, first, would be able to think recursively on the same level of embedding in different areas of application; and second, the general level of recursive thinking would, at least partly, predict her performance in the TB task.

## Rationale of the present study

In sum, the puzzling developmental pattern of the U-shaped performance curve in the TB task raises two fundamental questions: First, how does the decrease in performance in the TB task come about at the beginning of the U-shaped curve? Second, how does the recovery of performance come about at the end of the U-shaped curve? The pragmatic analysis presents one possible answer to both questions: the U-shaped curve reflects an underlying development in children's understanding of pragmatics.

Previous research has yielded some evidence that speaks to the first question, but so far, no study has empirically addressed the second question. The rationale of the present study, therefore, is to test whether, indeed, the developmental change at the end of the U-shaped performance curve reflects and can be explained by pragmatic development.

To this end, we tested a wide age range of children with the standard FB and TB task. Due to the restrictions of the Covid-19 pandemic, testing was conducted in an online format. We ran the studies as moderated online studies in which the experimenter interacted with the child via video chat while presenting the tasks on screen. With this change in setting, the pragmatic context in which the experimenter administers the TB task to the child was essentially different compared to earlier studies. A preliminary question thus was whether the typical performance curve replicates in the new format in children between six and ten years. The age range is expected to include younger children who still fail and older children who succeed again in the TB task (e.g., [6]), representing the right half of the U-shaped performance curve.

The main research question then was: What are the factors that explain the end of the U-shaped performance curve in the TB task? Here, we explore different possibilities based on the pragmatic analysis introduced above:

1. Is the TB pattern a function of advanced pragmatics development?

2. Even more generally: Is it a function of recursive ToM, or even of recursive thinking in general?

In order to do so, *Advanced Pragmatic Understanding* was operationalized in a task that asked for children's metaphor and irony understanding. *Recursive ToM* was operationalized in tasks that tested children's understanding and production of higher-order mental state ascriptions. The *general ability for recursive thinking* was operationalized as children's recursive language abilities as a proxy for their general recursive abilities. This task tested children's understanding for embedded recursive clauses. Children were tested in these tasks and the TB task in three online studies. Table 1 displays the tasks administered in Study 1–3.

## Study 1

### Method

This research was conducted in accordance with the Declaration of Helsinki and the Ethical Principles of the German Psychological Society (DGPs), the Association of German Professional Psychologists (BDP), and the American Psychological Association (APA). It involved no invasive or otherwise ethically problematic techniques and no deception (and therefore, according to National jurisdiction, did not require a separate vote by a local Institutional Review Board; see the regulations on freedom of research in the German Constitution (§ 5 (3)), and the German University Law (§ 22)). Before the test sessions of Studies 1–3 started, informed consent was obtained from the parents of the subjects.

**Table 1. Tasks included in Studies 1–3 in the order presented in the test session.**

| Task | Study 1 | Study 2 | Study 3 |
|---|---|---|---|
| Recursion in general: recursive language abilities | x | x | x |
| Change-of-location: TB and FB | x | x | x |
| Recursive ToM: production | | x | |
| Recursive ToM: understanding | x | x | x |
| Advanced pragmatics understanding | x | | |

**Design.** Children in all three studies were tested in a single session (30–45 minutes) by a female experimenter (E). The tasks were presented remotely (on a laptop computer screen or tablet computer screen, no smartphone) in an interactive online study via a video conferencing platform (mainly *BigBlueButton*, in case of connection issues, the test session was shifted to *Zoom*). The tasks were embedded in a video, which was displayed via the conference platform in the middle of the child's screen and required the child to give verbal answers. Next to the video, the child was constantly able to see the webcam video of E and herself, so that the child and E were able to communicate via audio and video streaming during the whole test session (*see* [7] for a validation study of this paradigm). Before the beginning of the test session, the caretaker gave verbal consent to the child's participation in the study and the video and audio recording during the test session. The verbal consent was recorded and stored separately from the recording of the test session. The caretaker and the child were informed that they might abort the participation at any given moment. In the beginning of the test session, E advised the child that she could repeat each question if the child had any comprehension difficulties.

**Participants.** Eighty-one 6- to 10-year-old children (72–131 months, *mean age* = 99.52 months; 41 girls, 40 boys) were included in the final sample. Eight additional children were tested but excluded from data analyses because of technical issues during the test session ($N = 6$), uncooperative behavior ($N = 1$) and concentration deficit resulting in >50% incorrectly answered control question in the change-of-location task ($N = 1$). The age range was chosen so broadly in order to compare children who show and do not show the performance difficulties in the TB task. Participants in this and all subsequent studies were recruited from a database of children whose parents had previously given consent to experimental participation as well as via social media.

**Material.** *Test for syntactic recursion*. The task adapted from Arslan and colleagues (2017) tested for the comprehension of embedded relative clauses in German language [39]. Children saw two rows of animals (upper and lower row) on the screen (Fig 1). Each animal was displayed on a different background color. The children were asked to name the location of a corresponding animal on the screen (e.g., "Where is the cow that strokes a horse?" for the first-order syntactic recursion test question). Children had to refer to the animal's location in naming the corresponding background color and the row in which the animal was placed (e.g., "yellow, upper row"). The test questions containing the relative clauses were scaled from first order until fourth order of syntactic recursion and could be repeated up to four times [39]. For a detailed procedure, *see* S1 File.

*Standard change-of-location task*. The children received four trials of the standard change-of-location tasks with different stimuli [2] implemented in short, animated video clips. Protagonist A and her object O were presented to the child before Protagonist A placed O in one of two boxes (box 1). In her presence (TB condition) or absence (FB condition), protagonist B came into the scene and moved O to the other box (box 2) and the following test and control questions were asked:

- Test question: "Where does Protagonist A think that O is?" [correct answer box 1 for FB, box 2 for TB condition]

- Control Question 1: "In which box was O in the beginning?" [correct answer: box 1]

- Control Question 2: "Where is O now?" [correct answer: box 2]

The TB and FB trials were presented in alternating order beginning with a TB trial and the children saw a frozen still image of the last frame of the scenario (Protagonist A and the two boxes) when answering the questions.

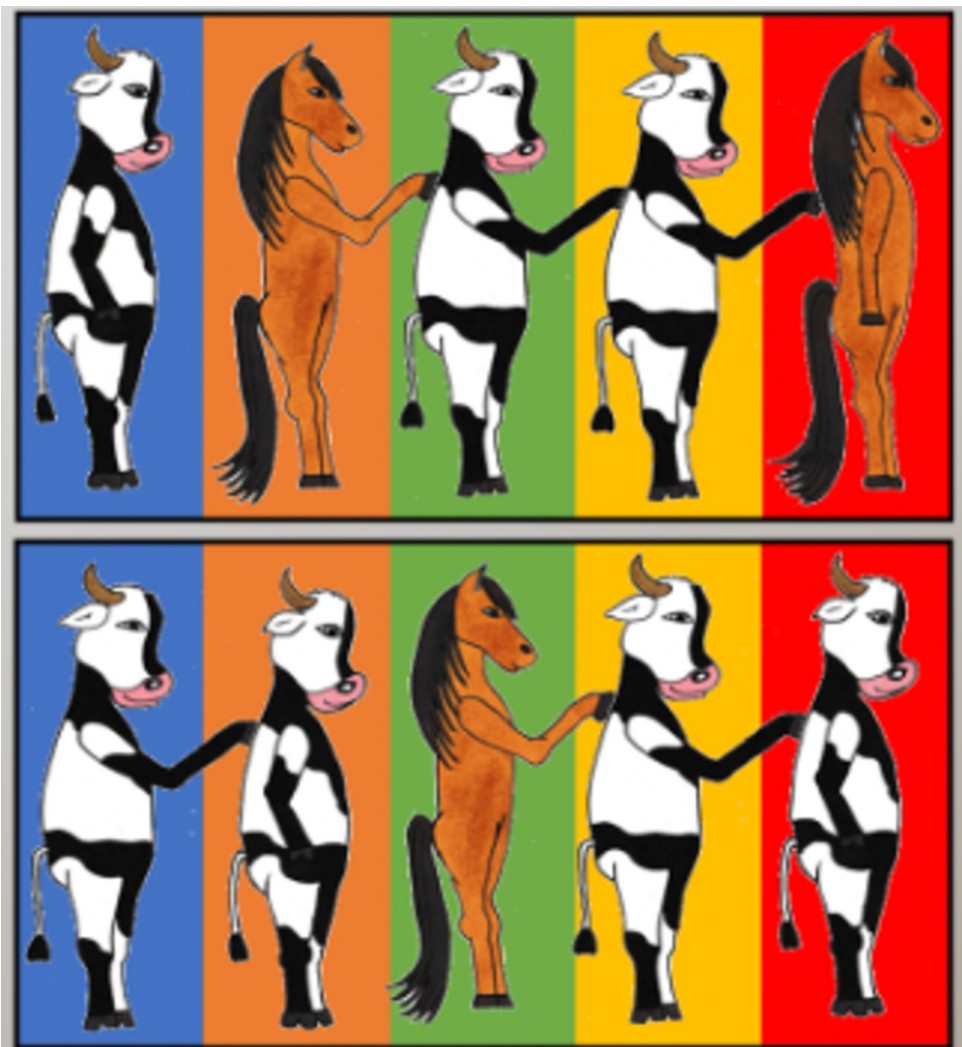

**Fig 1. Material for test question with a first-order syntactic recursion "Where is the cow that strokes a horse?", Correct answer "yellow, upper row".**

*Recursive ToM task*: *Understanding*. The children heard three stories (partly adapted from [33]) accompanied by animated video clips and were asked to answer test questions about the characters' mental states afterwards. The test questions were scaled from second order to fifth order of mental state recursion and children had to decide which of two sentences was true regarding the story line. The sentences were read out by a voice and displayed with pictures on the screen (Fig 2). To choose one of the two sentences, children could either name the side/ color of the picture on the screen or repeat the sentence.

Example story: the video dilemma (adapted from [33])

This is Sarah and this is Olli. Sarah and Olli are in the same class at school. "Hi, I'm Sarah!', "and I'm Olli". Their teacher is Mrs. Brown. Today Mrs. Brown suggests that Sarah and Olli should bring a video to school tomorrow to watch with the other children. Mrs. Brown also says to them, "Make sure you bring a film that I will like too!" (*Mrs. Brown leaves the scene*). Sarah's favorite videos are pirate videos. Olli's favorite videos are horse films. Which will it be? A pirates or a horse film? Olli says to Sarah, "We just can't decide so I think that we should

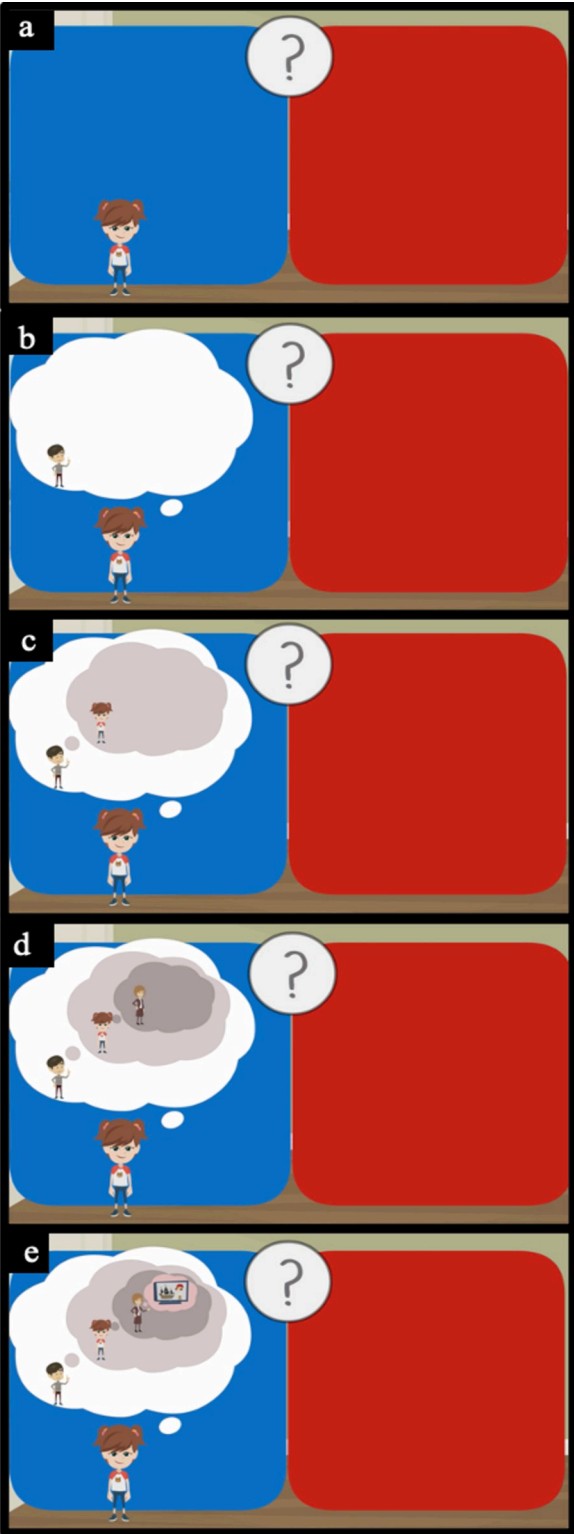

**Fig 2. Screenshots of the visual animation of the fourth-order Theory of Mind test question in the story line "the video dilemma".** *Note.* Children heard a voice slowly reading out the answer sentences while the animation was presented accordingly on the screen. E.g., "Sarah hopes [picture a] that Olli believes [picture b] that she knows [picture c] that Mrs. Brown wants [picture d] them to watch a pirate film [picture e]". After that the second answer option (red side) was read out and presented accordingly.

take the film that Mrs. Brown would like. Sarah, do you know which one Mrs. Brown would like best?" Sarah is thinking about that. She does not have a clue which film Mrs. Brown would like. But Sarah decides to tell Olli that she knows that Mrs. Brown likes pirate films best. Sarah thinks that this will make Olli agree to take a pirate video to school. Olli listens to this and then Olli says, "We will take a video of pirates then." So, Sarah gets to enjoy her favorite film!

Memory question

Which sentence is true?

a) Sarah likes pirates films best.

b) Sarah likes horse films best.

Test question (ToM Level 4)

Which sentence is true?

a) Sarah hopes that Olli believes that she knows that Mrs. Brown wants that they watch a pirate film.*

b) Sarah hopes that Olli believes that she doesn't know that Mrs. Brown wants them to watch a pirate film.

*German translation with that-complement for want ("möchte, dass")

*Task for advanced pragmatics understanding.* Children received two trials of a pragmatic language task testing for their metaphor and irony understanding [partly adapted from and inspired by 20, 26, 30, 31]. Each trial consisted of a story about two characters accompanied by three pictures (Table 2 and example below).

The questions (and answer options) could be repeated up to four times. During the ironic utterance (Table 2, Picture 3), the speaker's face was not visible to avoid any inferences from their facial expression. To answer the second test question correctly, children had to refer to the speaker's mental state or attitude to the other agent's behavior or refer to the negative outcome of the other agent's behavior or to the opposite/ ironic meaning of the utterance. Answers to this test questions were coded with a fixed coding scheme (adapted from [25, 26], *see* S1 File).

## Results

**Coding of predictors.** In the syntactic recursion task and the recursive ToM understanding task, children were coded with the highest level of recursion until that they performed consistently correct (e.g., child is coded with "3" when she answers the test questions for level 1–3

**Table 2. Example story for Advanced Pragmatics Understanding Task.**

| Picture | Content | Example |
|---|---|---|
| 1 | Information for metaphor | Lisa is running through the apartment all day. She plays with the ball, jumps on the sofa and plays tag with the cat. |
| | Test Question about metaphor | What fits the best? Lisa is. . .<br>• A cloud<br>• A crocodile<br>• A whirlwind (Correct Answer: Metaphor used in German for very active children)<br>• A tree |
| 2 | Story line | In this moment, Lisa is running so fast that she hits the table and all the books fall on the ground. |
| 3 | Ironic utterance | Lisa's older brother enters the rooms and says, "You're very careful today." |
| | Questions about ironic utterance | Test question 1: Does the brother want Lisa to believe that he thinks that she was careful? |
| | | Test question 2: Why does the brother say, "You're very careful today"? |
| | | Control question: Does the brother find that Lisa was careful? |

correctly, but the test question for level 4 incorrectly). For advanced pragmatics, children received the score of correct trials for the metaphor test questions, the irony test question 1 and irony test question 2 separately (0–2 each). For the coding scheme for irony test question 2 and interrater reliabilities, *see* S1 File.

**Plan of analysis.** In a first step, we assured that the children responded consistently in the two trials of the same condition in the change-of-location task, so that we were able to code children's performance in this task for the subsequent analysis in a binary format (passers vs. non-passers).

Second, in scope of the preliminary analyses, we tested for the typical performance in the TB and FB task in computing comparisons against chance level performance for both TB and FB in the three age groups (young, middle, old). Children of all groups were expected to perform above chance level in the FB task whereas only the oldest age group (9;4–10;11 years) was expected to perform better than chance level performance in the TB task. Additionally, we computed correlations between FB and TB performance which were expected to be negative for the two younger age groups and positive for the oldest age group only.

To address the main research question of factors that influence the performance in the TB task, we computed a logistic regression model. In the logistic regression model, TB performance (passing vs. no-passing) was predicted by recursive syntactic abilities, recursive ToM understanding, advanced pragmatics understanding and children's age. We compared this full model with a control model containing only children's age in months.

**TB and FB performance: Consistency across trials in the standard change-of-location task.** The consistencies in performance of children over the two trials of the same condition of the standard change-of-location task were high. The percentage of children who had two available trials (meaning all control questions answered correctly) and showed the same performance in both trials was 85.90% ($\Phi$ = .68) for the TB trials and 98.75% ($\Phi$ = n/c, due to at least one constant variable) for the FB trials. Therefore, both trials were included in the analysis. For the following analysis, the TB and FB performance were coded as binary variables. Children had to pass both trials of a condition to be assigned to the group of passers. Children failing in one or both trials of a condition were assigned to the group of non-passers.

**Preliminary analyses: TB and FB performances in different age groups of children.** The performance in the change-of-location task as a function of belief type and age is depicted in Fig 3.

To test for the failure in the TB condition and the success in the FB condition of younger children and the success in both conditions in older children, we computed Wilcoxon signed rank tests against chance level performance (0.5) for the three age groups and the two belief conditions. The Wilcoxon tests showed that the youngest age group (6;0–7;7-year-olds) performed significantly above chance in the FB condition ($M$ = .93, p < .0001, r = -.85). The tests could not be computed for the two older age groups due to ceiling effects in the FB condition (7;8–9;3-year-olds and 9;4–10;11-year-olds $M$ = 1). In contrast, Wilcoxon signed rank tests revealed that in the TB condition, only the oldest age group of children (9;4–10;11-year-olds) performed significantly above chance ($M$ = .74, $p$ < .02, $r$ = —.48). Younger children (6;0–7;7-year-olds and 7;8–9;3-year-olds) performed at chance level (6;0–7;7-year-olds: $M$ = .41, $p$ = .34, $r$ = —.18; 7;8–9;3-year-olds: $M$ = .63, $p$ = .18, $r$ = -.26).

The correlation between the TB and FB performance in the change-of-location task is (not-significantly) negative for the whole sample (r(phi) = -.13, $p$ = .24) as well as for the youngest age group (6;0–7;7-year-olds: $r$(phi) = -.34, $p$ = .08). Because of the ceiling effects in the FB condition, the correlation is not computable for the two older age groups.

**Main analyses: Predictors for TB performance.** We removed children failing the first-order FB condition ($N$ = 2) from the following analyses. This was based on the assumption

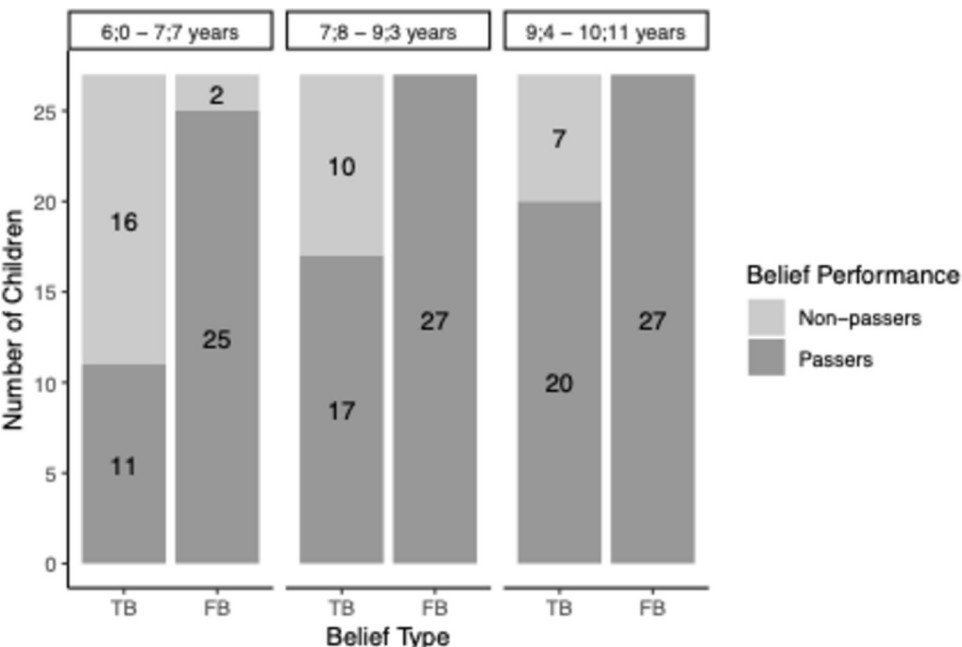

**Fig 3. Children's performances in the standard change-of-location task as a function of age group and belief type in Study 1.**

that children who still do not succeed in the first-order FB task use different cognitive strategies to solve the TB task compared to the group of children we aim to examine here.

*Descriptive statistics.* The mean performances in the recursive ToM understanding task, the advanced pragmatic language task and the syntactic recursion task as a function of TB performance and age are summarized in Table 3.

For a more detailed summary of answers to irony test question 2, *see* S1 File.

*Logistic regression models.* We estimated the effect of the different predictors of mental state ascription on the TB performance using a multiple logistic regression model. To control for children's age in months, we included it into the model, too. Prior to fitting the model, we checked for the assumptions. We checked for multicollinearity (all *VIF*s ≤ 1.38) and linearity of the logit for age ($b = 0.08$, $p = .81$), recursive ToM understanding ($b = 1.48$, $p = .73$) and syntactic recursion ($b = -2.98$, $p = .27$).

We compared the fit of the full model with that of a null model with the control variable only (TB ~ age). As the model comparison is significant, the predictors of mental state

**Table 3. Mean performance (M) and standard deviations (SD) in Syntactic Recursion (Synt. Recurs.), Recursive ToM Understanding (RToM U), metaphor understanding (Metaphor), irony understanding in the first and second test question (Irony1 and Irony2) and for TB non-passers (noTB) and TB passers (TB) in three groups of age.**

| | n | | Synt. Recurs. | | RToM U | | Metaphor | | Irony1 | | Irony2 | |
|---|---|---|---|---|---|---|---|---|---|---|---|---|
| | | | M(SD) | | M(SD) | | M(SD) | | M(SD) | | M(SD) | |
| Age | noTB | TB | noTB | TB | noTB | TB | noTB | TB | noTB | TB | noTB | TB |
| 6;0–7;7 | 16 | 11 | 2.19 (1.11) | 2.00 (1.5) | 2.13 (1.36) | 2.44 (0.88) | 1.33 (0.62) | 1.78 (0.67) | 1.44 (0.81) | 1.00 (1.00) | 1.40 (0.83) | 0.89 (0.93) |
| 7;8–9;3 | 10 | 17 | 2.80 (0.92) | 2.94 (1.25) | 2.00 (0.67) | 4.18 (1.19) | 1.80 (0.42) | 1.82 (0.39) | 1.00 (0.94) | 1.35 (0.79) | 1.40 (0.84) | 1.24 (0.83) |
| 9;4–10;11 | 7 | 20 | 2.43 (1.62) | 3.00 (0.97) | 2.57 (1.40) | 3.90 (1.37) | 1.71 (0.49) | 1.80 (0.41) | 1.71 (0.49) | 1.55 (0.60) | 1.86 (0.38) | 1.75 (0.44) |

*Note.* Possible range of performances for recursive ToM understanding: 0–5, for pragmatic language task: 0–2, for syntactic recursion task: 0–4.

**Table 4. Results of the logistic regression model predicting children's TB performance with their age in months and their performance in tasks of syntactic recursion (Synt. Recurs.), Recursive ToM understanding (RToM U), and Advanced Pragmatics (Metaphor and Irony1 and Irony2 for irony test questions 1 and 2).**

| | B(SE) | z | p | 95% CI for Odds Ratio | | |
| --- | --- | --- | --- | --- | --- | --- |
| | | | | Lower | Odds Ratio | Upper |
| Included | | | | | | |
| Constant | -4.10 (1.89) | -2.17 | .03* | | | |
| Age in months | 0.02 (0.02) | 1.29 | .20 | 0.99 | 1.02 | 1.06 |
| Synt. Recurs. | -0.19 (0.29) | -0.66 | .51 | 0.46 | 0.83 | 1.44 |
| RToM U | 0.84 (0.23) | 3.67 | < .001*** | 1.54 | 2.33 | 3.84 |
| Metaphor | 0.71 (0.60) | 1.18 | .24 | 0.64 | 2.03 | 6.95 |
| Irony1 | 0.13 (0.39) | 0.34 | .73 | 0.52 | 1.14 | 2.48 |
| Irony2 | -0.68 (0.43) | -1.59 | .11 | 0.21 | 0.51 | 1.15 |

*Note.* $R^2$ = .44 (Nagelkerke). Model $X^2(6)$ = 24.58, $p < .001$.

***$p < .001$.

ascription have an impact on the TB performance (Model $X^2(6)$ = 24.58, $p < .001$). More specifically, an increased ability in recursive ToM understanding lead to increased TB performance (B = 0.84, $p < .001$***, OR = 2.33). Pragmatic language abilities, age and syntactic recursion abilities did not affect the TB performance significantly (Table 4).

Fig 4 pictures this difference in performance in recursive ToM understanding between TB-passers and TB-non-passers.

*Post-hoc analyses.* We computed post-hoc one-sided Wilcoxon rank sum tests for TB-passers versus TB-non-passers in recursive ToM understanding as it significantly predicted the outcome in the logistic regression model. Due to multiple testing, Bonferroni correction was applied and resulted in an alpha value of 0.0125 (0.05/4) for this post-hoc computation. The comparison of the performance shows a significant difference in the recursive ToM

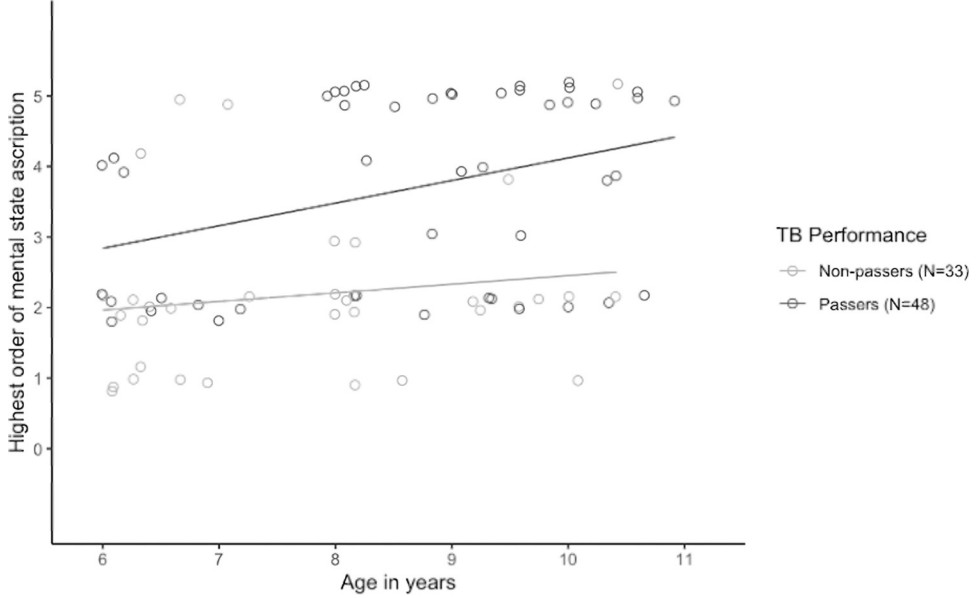

**Fig 4. Children's performance in recursive ToM understanding as a function of their TB performance (passers vs. non-passers) across age.**

understanding for TB-non-passers ($M$ = 1.91, $n$ = 33) against TB-passers ($M$ = 3.72, $n$ = 46, $W$ = 319.5, $p < .0001$, $r$ = -.53) for the whole sample. For the comparison within the age groups, Wilcoxon rank sum tests reveal that the performance differs significantly for the middle age group between TB-non-passers ($M$ = 1.80, $n$ = 10) and TB-passers ($M$ = 4.18, $n$ = 17, $W$ = 16, $p < .0001$, $r$ = -.73). This difference is not significant for the youngest age group ($M$ (noTB) = 1.75, $n$ = 16; $M$(TB) = 2.44, $n$ = 9, $W$ = 50.5, $p$ = .09, $r$ = -.34) and the oldest age group of children ($M$(noTB) = 2.43, $n$ = 7; $M$(TB) = 3.90, $n$ = 20, $W$ = 34.5, $p$ = .02, $r$ = -.45).

The achieved power was computed post-hoc for the logistic regression model. For the significant predictor recursive ToM understanding, this resulted in a power of $1 - \beta$ = .89.

## Discussion

The expected pattern of typical TB and FB performance in children between six and ten years was replicated in the online study: children in the youngest and middle age group (6;0–9;3 years) failed to perform above chance level in the TB task while FB performance was at ceiling. Only children in the oldest age group (9;4–10;11years) performed proficiently in both conditions. Additionally, the study shows first evidence that children's TB performance can be (partly) explained by their understanding for recursive ToM. However, none of the variables of Advanced Pragmatics understanding or syntactic recursion were significant predictors for children's TB performance.

## Study 2

Study 2, therefore, aimed to replicate this relation between TB performance and children's recursive ToM. In order to explore the underlying recursive ToM abilities in more detail, Study 2 operationalized children's recursive ToM twofold: similar to Study 1, children's understanding for recursive mental state ascriptions was measured. Additionally, children's recursive ToM production was measured to identify the cognitive mechanisms relevant for the TB task more fine-grained.

### Method

**Participants.** The final sample included eighty-seven 6- to 10-year-old children (72–131 months, *mean age* = 101.41 months; 44 girls, 43 boys). Seven additional children were tested but excluded from data analyses because of technical issues during the test session ($N$ = 3), experiential error ($N$ = 1), uncooperative behavior ($N$ = 1), parental interference ($N$ = 1) or children's age (child turned out to be too old on the day of the test session; $N$ = 1).

**Material.** *Test for syntactic recursion*. The task for syntactic recursion was administered as in Study 1.

*Standard change-of-location task*. For the standard change-of-location task, we used the same material as in Study 1. As the consistency of the two trials (two FB and two TB trials) in Study 1 was high, only one trial per condition was administered in Study 2. Again, the TB trial was always presented first.

*Recursive ToM production*. Children saw two story lines containing bluffs [partly adapted from [40]]. In the first story, a character bluffed to mislead someone. In the second story line, the character first uttered a double bluff, i.e. he told the truth as the opponent expected him to lie. As the second story line continued, the same character again uttered a bluff based on his double bluff: he lied as the opponent now expected him to say the truth (triple bluff). The story lines were presented as short animated video clips. After each utterance, children were asked to explain the bluffs ("Why does he say that?"). To receive detailed answers, E asked the child to explain her answer in more detail. The exact wording of this follow-up question depended

on the length of the child's initial answer. Answers to the open question were coded binary (child did explain the bluff correctly (1) or did not understand the bluff (0)). Children received the highest score of bluffs that were all explained correctly (0–3).

Example of task for recursive ToM production

Double bluff [shortened, adapted from 40]

The treasure diggers want to find the pirates' treasure. They know that the treasure is either in the field or in the mountains. They hold one of the pirates captive and ask him where the treasure is. The captured pirate is very brave and very smart, he will not let the treasure diggers find the treasure. The treasure is in the mountains. When the treasure diggers ask the pirate where the treasure is, the pirate answers, "in the mountains".

Control question: Is that right what he said? (correct answer: yes)

Test Question: Why did he say that?

For follow-up test question depending on length of the answer to initial test question, *see* S1 File.

*Recursive Tom*: *Understanding*. The task testing for the understanding of recursive ToM was taken over from Study 1 and extended: Children received one additional story and four additional test questions resulting in four story lines, two test questions per story and order of recursion (for more details, *see* S1 File). If a child answered both second order questions or more than 50% of the first four test questions incorrect, the test session was terminated after the third story line.

## Results

**Coding of predictors.** In the syntactic recursion task and the recursive ToM understanding and production task, children were coded with the highest level of recursion up to which they answered the test question/ both test question correctly (e.g., child is coded with "3" for recursive ToM understanding when she answers all test questions for level 1–3 correctly, one test question for level 4 correctly and one test question for level 4 incorrectly). For recursive ToM production, children receive the score of the respective bluff they explained correctly (0–3; "1" when they only explained the bluff correctly, "2" when they explained the bluff and the double-bluff correctly and "3" when they explained all three bluffs correctly). For interrater reliabilities, *see* S1 File.

**Plan of analysis.** The analysis was conducted in close similarity to Study 1. In scope of the preliminary analyses, we tested for the typical performance in the TB and FB task in computing comparisons against chance level performance for both TB and FB in the same three age groups (young, middle, old) and computed correlations between FB and TB performance.

To address the main research question of factors that influence the performance in the TB task, we again computed a logistic regression model. In the logistic regression model, TB performance (passing vs. no-passing) was predicted by children's age, recursive syntactic abilities, recursive ToM production and recursive ToM understanding. We compared this full model with a control model containing only children's age.

**Preliminary analyses: TB and FB performances in different age groups of children.** The performance in the change-of-location task as a function of belief type and age is depicted in Fig 5.

To test for the failure in the TB condition and the success in the FB condition of younger children and the success in both conditions in older children, we computed Wilcoxon signed rank tests against chance level performance (0.5) for the three age groups and the two belief conditions. Wilcoxon showed that the youngest age group (6;0–7;7-year-olds) performed significantly above chance in the FB condition ($M = .86$, $p < .001$, $r = -.70$). Wilcoxon tests could

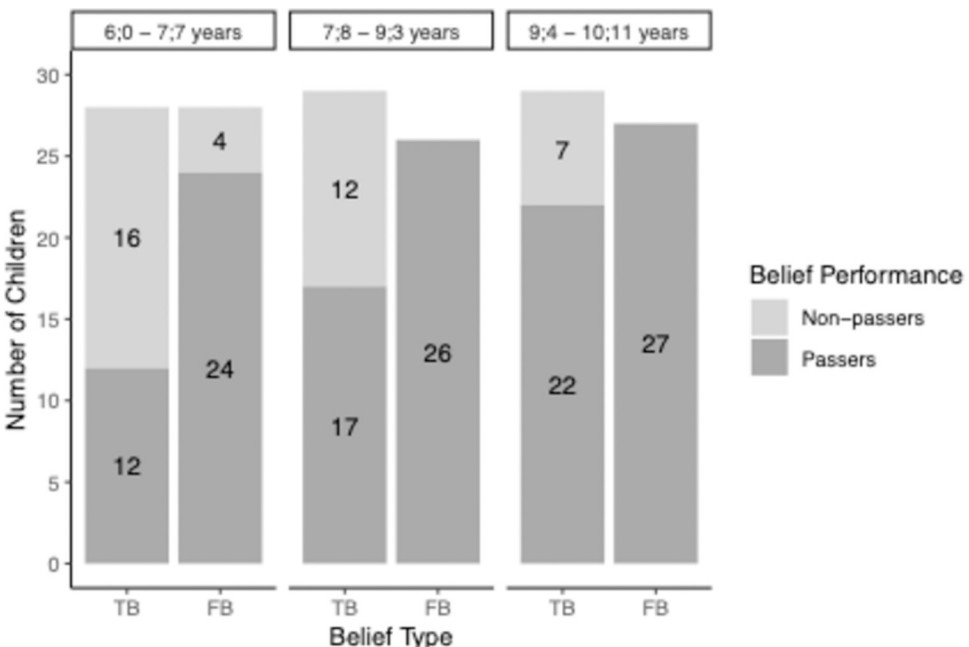

**Fig 5. Children's performance in the change-of-location task as a function of belief type and age in Study 2.** *Note.* Sample size of the groups vary as only children were included who answered the respective control questions correctly.

not be computed for the two older age groups due to ceiling effects in the FB condition (7;8–9;3-years-olds and 9;4–10;11-year-olds: $M = 1$). In contrast, Wilcoxon signed rank tests revealed that in the TB condition, only the oldest age group of children (9;4–10;11-year-olds) performed significantly above chance ($M = .76$, $p < .01$, $r = -.51$). Younger children (6;0–7;7-year-olds and 7;8–9;3-year-olds) performed at chance level (6;0–7;7-year-olds: $M = .43$, $p = .46$, r = -.14, 7;8–9;3-year-olds: $M = .59$, $p = .36$, $r = -0.17$).

The correlation between the TB and FB performance in the change-of-location task is (not-significantly) negative for the whole sample ($r$(phi) = -.19, $p = .10$) and significantly negative for the youngest age group (6;0–7;7-year-olds: $r$(phi) = -.47, $p = .02$). Because of the ceiling effects in the FB condition, the correlation is not computable for the two older age groups.

**Main analyses: Predictors for TB performance.** *Descriptive statistics.* The mean performances in the recursive ToM understanding task, the recursive ToM production task and the syntactic recursion task as a function of TB performance (passers vs. non-passers) and age (young, middle, old age group) are summarized in Table 5.

**Table 5. Mean performance (M) and standard deviations (SD) in Recursive ToM Understanding (RToM U), Recursive ToM Production (RToM P) and understanding of syntactic recursion (Syntact Recurs.) for TB non-passers and TB passers in three groups of age.**

|  | *n* |  | Synt. Recurs. |  | RToM P |  | RToM U |  |
|---|---|---|---|---|---|---|---|---|
|  |  |  | *M(SD)* |  | *M(SD)* |  | *M(SD)* |  |
| Age | noTB | TB | noTB | TB | noTB | TB | noTB | TB |
| 6;0–7;7 | 16 | 12 | 2.19 (1.11) | 2.08 (1.16) | 1.06 (0.25) | 1.17 (0.39) | 1.25 (1.06) | 1.17 (1.03) |
| 7;8–9;3 | 12 | 17 | 3.00 (0.95) | 3.00 (1.12) | 1.67 (0.89) | 2.18 (1.07) | 2.92 (0.90) | 2.65 (1.46) |
| 9;4–10;11 | 7 | 22 | 2.57 (1.27) | 3.00 (1.02) | 1.86 (0.90) | 1.95 (0.90) | 2.71 (1.60) | 3.36 (1.43) |
| all | 35 | 51 | 2.54 (1.12) | 2.78 (1.14) | 1.43 (0.74) | 1.84 (0.95) | 2.11 (1.37) | 2.61 (1.59) |

*Note.* Possible range of performance for Recursive ToM Understanding task: 0–5, for Recursive ToM Production: 0–3, for Syntactic Recursion task: 0–4.

**Table 6. Results of the logistic regression model predicting children's TB performance with their age in months and their performance in tasks of syntactic recursion (Synt. Recurs.), Recursive ToM understanding (RToM U), and Recursive ToM Production (RToM P).**

| | *B(SE)* | *z* | *p* | 95% CI for Odds Ratio | | |
| | | | | Lower | Odds Ratio | Upper |
|---|---|---|---|---|---|---|
| Included | | | | | | |
| Constant | -3.09 (1.52) | -1.88 | .04 | | | |
| Age in months | 0.02 (0.02) | 1.04 | .15 | 0.99 | 1.02 | 1.06 |
| Synt. Recurs. | 0.12 (0.24) | 0.41 | .62 | 0.70 | 1.12 | 1.80 |
| RToM P | 0.26 (0.36) | 0.73 | .47 | 0.66 | 1.30 | 2.75 |
| RToM U | 0.11 (0.20) | 0.54 | .59 | 0.75 | 1.11 | 1.67 |

*Note.* $R^2$ = .16 (Nagelkerke). Model $X^2(3)$ = 2.95, $p$ = .40

*Logistic regression models.* We again removed children failing the first-order FB condition (or having no FB trial available due to incorrect answers to the control questions in the FB condition, $n$ = 10) from the following analyses. Prior to fitting the model, we checked for the assumptions. We checked for multicollinearity (all *VIFs* $\leq$ 1.49) and linearity of the logit for Age ($b$ = 0.11, $p$ = .61), recursive ToM understanding ($b$ = 1.32, $p$ = .43), recursive ToM production ($b$ = 1.32, $p$ = .43) and syntactic recursion ($b$ = 1.93, $p$ = .15).

None of the predictors in the full model predicted significantly children's TB performance (Table 6). The comparison of the full model with the null model containing the control variable only (TB ~ age) was not significant ($X^2(3)$ = 2.95, $p$ = .40).

*Post-hoc comparisons.* We did not compute post-hoc one-sided two-sample Wilcoxon tests for TB-passers versus TB-non-passers in recursive ToM as it did not significantly predict the outcome in the logistic regression model in Study 2.

## Discussion

Study 2 did not show the relationship between children's TB performance and their recursive ToM found in Study 1, neither operationalized in terms of their understanding for recursive ToM nor in their own production of recursive ToM. The same task for children's recursive ToM understanding was used as in Study 1, however, the task was extended by an additional storyline and additional test questions and the task for recursive ToM production was added. This increased the duration of testing significantly and may have had a negative outcome on children's concentration and motivation in the test session and, therefore, the validity of results.

## Study 3

We therefore conducted Study 3 in which the initial relationship between children's TB performance and their recursive ToM understanding was tested without additional tasks testing for Advanced Pragmatics or recursive ToM production to reduce the duration of the test session and hold children's concentration as constant as possible.

## Method

**Participants.** The final sample included sixty-four 6- to 10-year-old children (72–131 months, *mean age* = 102 months; 32 girls, 32 boys). Five additional children were tested but excluded from data analyses because of technical issues during the test session ($N$ = 4) and uncooperative behavior ($N$ = 1).

**Material.** The task for syntactic recursion, the standard change-of-location task and the task for recursive ToM understanding were administered with the same material and procedure as in Study 2. In contrast to Study 2, there was no termination rule for the task testing for recursive ToM understanding.

## Results

**Coding of predictors.** As in Study 2, children were coded with the highest level of recursion up to which they answered all test question correctly (that is, one test question in the syntactic recursion task and two test questions in the recursive ToM understanding task).

**Plan of analysis.** The analysis was conducted in close similarity to Study 1 and 2. In scope of the preliminary analyses, we again tested for the typical performance in the TB and FB task in computing comparisons against chance level performance for both TB and FB in the three age groups (young, middle, old) and computed correlations between FB and TB performance.

To address the main research question of factors that influence the performance in the TB task, we again computed a logistic regression model. In the logistic regression model, TB performance (passing vs. no-passing) was predicted by children's age, recursive syntactic abilities and recursive ToM understanding. We compared this full model with a control model containing only children's age.

**Preliminary analyses: TB and FB performances in different age groups of children.** The performance in the change-of-location task as a function of belief type and age is depicted in Fig 6. Children were included when they answered the respective control questions correctly.

To test for the failure in the TB condition and the success in the FB condition of younger children and the success in both conditions in older children, we computed two-sided Wilcoxon tests against chance level performance (0.5) for the three age groups and the two belief

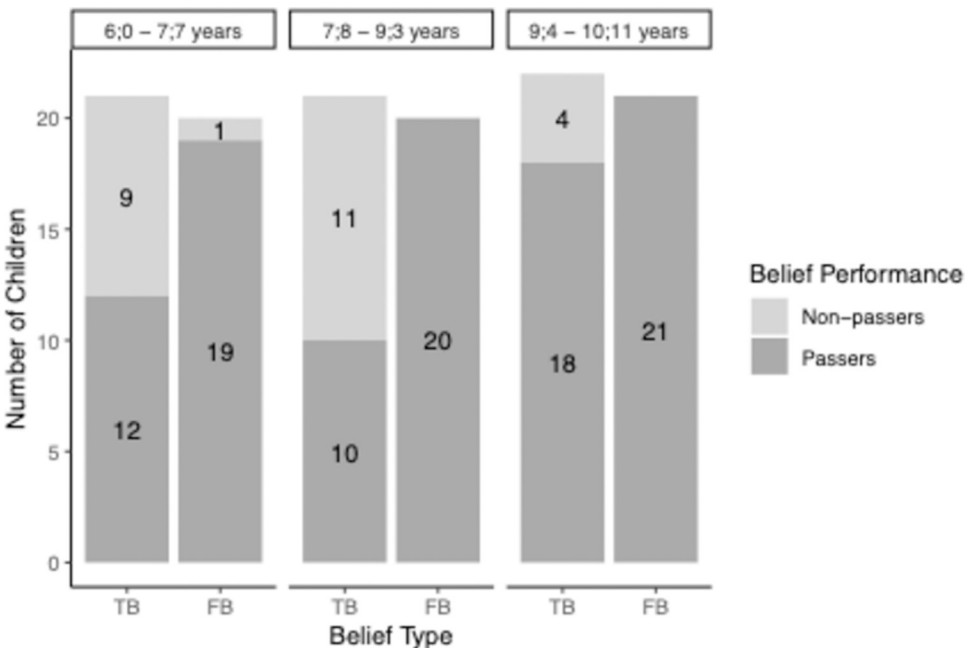

**Fig 6. Children's performance in the standard change-of-location task as a function of belief type and age in Study 3.** *Note.* Sample size of the groups vary as only children were included who answered the respective control questions correctly.

**Table 7. Mean performance (M) and standard deviations (SD) in understanding of syntactic recursion (Synt Recurs.) and Recursive ToM Understanding (RToM_U) for TB non-passers and TB passers in three groups of age.**

| | n | | Synt. Recurs. | | RToM U | |
| | | | M(SD) | | M(SD) | |
| Age | noTB | TB | noTB | TB | noTB | TB |
|-----|------|-----|--------|------|-------|------|
| 6;0–7;7 | 9 | 12 | 2.11 (1.17) | 2.83 (0.83) | 2.00 (0.50) | 1.25 (0.62) |
| 7;8–9;3 | 11 | 10 | 2.55 (1.21) | 3.00 (0.94) | 1.73 (1.01) | 2.50 (1.08) |
| 9;4–10;11 | 4 | 18 | 3.00 (1.15) | 3.11 (1.28) | 2.75 (1.71) | 3.50 (1.42) |
| all | 24 | 40 | 2.46 (1.18) | 3.00 (1.06) | 2.00 (1.02) | 2.58 (1.48) |

*Note*. Possible range of performances for syntactic recursion task: 0–4, for Recursive ToM Understanding task: 0–5.

conditions. The Wilcoxon tests showed that the youngest age group (6;0–7;7-year-olds) performed significantly above chance in the FB condition ($M = .95$, $p < .0001$, $r = -.87$). Wilcoxon tests could not be computed for the two older age groups due to ceiling effects in the FB condition (7;8–9;3-year-olds and 9;4–10;11-year-olds $M = 1$). In contrast, Wilcoxon tests revealed that in the TB condition, only the oldest age group of children (9;4–10;11-year-olds) performed significantly above chance ($M = .82$, $p < .01$, $r = -.65$). Younger children performed at chance level (6;0–7;7-year-olds: $M = .57$, $p = .53$, r = -.14, 7;8–9;2-year-olds: $M = .48$, $p = .84$, $r = -.04$).

The correlation between the TB and FB performance in the change-of-location task is (non-significantly) negative for the whole sample ($r$(phi) = -.10, $p = .46$) and (non-significantly) negative for the youngest age group (6;0–7;7-year-olds: $r$(phi) = -.19, $p = .43$). Because of the ceiling effects in the false belief condition, the correlation is not computable for the two older age groups.

**Main analyses: Predictors for TB performance.** *Descriptive statistics*. The mean performances in the syntactic recursion task and the recursive ToM understanding task as a function of TB performance (passers vs. non-passers) and age (young, middle, old age group) are summarized in Table 7.

*Logistic regression models*. We again removed children failing to answer first-order FB test and/or control questions correctly ($n = 4$) from the following analyses. Prior to fitting the model, we checked for the assumptions. We checked for multicollinearity (all $VIFs \leq 1.55$) and linearity of the logit for age ($b = 0.19$, $p = .40$), syntactic recursion ($b = -2.42$, $p = .11$) and recursive ToM understanding ($b = 1.61$, $p = .18$).

None of the predictors in the full model predicted significantly children's TB performance (Table 8). The comparison of the fit of the full model with the null model with the control variable only (TB ~ age) was not significant ($X^2(1) = 0.07$, $p = .79$).

**Table 8. Results of the logistic regression model predicting children's TB performance with their age in months and their performance in tasks of syntactic recursion (Synt. Recurs.) and Recursive ToM understanding (RToM U).**

| | | | | 95% CI for Odds Ratio | | |
| | B(SE) | z | p | Lower | Odds Ratio | Upper |
|---|-------|---|---|-------|-----------|-------|
| Included | | | | | | |
| Constant | -2.80 (1.80) | -1.56 | .12 | | | |
| Age in months | 0.02 (0.02) | 1.23 | .26 | 0.98 | 1.02 | 1.06 |
| Synt R | 0.34 (0.26) | 1.31 | .19 | 0.85 | 1.40 | 2.37 |
| RToM U | 0.08 (0.29) | 0.26 | .79 | 0.61 | 1.08 | 1.93 |

Note. $R^2 = .12$ (Nagelkerke). Model $X^2(1) = 0.07$, $p = .13$.

**Table 9. Partial correlation of predictors in Study 1–3 controlled for children's age in months.**

| | Study 1 | | | | | Study 2 | | Study 3 |
|---|---|---|---|---|---|---|---|---|
| | 1 | 2 | 3 | 4 | 5 | 1 | 2 | 1 |
| 1. Synt | | | | | | | | |
| 2. RToM P | | | | | | .27** | | |
| 3. RToM U | .30** | | | | | .07 | .11 | .32** |
| 4. Meta | .15 | | .15 | | | | | |
| 5. Irony1 | -.16 | | -.04 | -.14 | | | | |
| 6. Irony2 | .20 | | .00 | .01 | .37*** | | | |

** indicates $p < .01$

*** indicates $p < .001$.

## Discussion

Similar to Study 2, the present study shows the typical TB performance in children between six and ten years but fails to replicate the relationship between children TB performance and their recursive ToM understanding that was found in Study 1. As neither children's syntactic recursion abilities nor their recursive ToM understanding predicted their TB performance, the results of the study therefore do not match with the predictions made by the pragmatic performance analysis.

**Supplementary explorative analysis: Correlations of predictors in Study 1–3.** In addition to the planed analysis, we conducted an exploratory analysis to investigate how the various predictors relate to each other. To this end, we computed partial correlations for the predictors in each study controlled for children's age in months (Table 9).

Table 9 shows a medium correlation between children's performance in the syntactic recursion task and recursive ToM understanding task for Study 1 and 3, but not in study 2. In study 2, however, children's performance in the syntactic recursion and recursive ToM production task correlated significantly. Additionally, children's performance in the first and second irony test question of the advanced pragmatics task in Study 1 show a medium to high correlation (Table 9).

## General discussion

Background to the present study was a puzzling empirical phenomenon: studies that administered the TB version of the classical change-of-location task to a broad age range of children yielded a surprising U-shaped performance curve: young children master this task, then from around age four children come to fail and performance only recovers around age eight to ten. Based on evidence that suggests that the decline in performance around age four reflects pragmatic confusions, the current set of studies tested whether performance recovery at the end of the U-shaped performance curve can be explained by another developmental change in children's pragmatic understanding. The studies therefore aimed to replicate the typical TB pattern (in an online testing format) and addressed potential factors that might explain the end of the U-shaped curve in two ways:

1. Is the TB pattern a function of advanced pragmatic development?

2. Even more generally: Is it a function of recursive ToM or recursive thinking in general?

The results, first of all, replicate the typical TB pattern in children between six and ten years: only the oldest age group (9;4–10;11 years) answered the TB task correctly while

younger children performed at chance level. Investigating potential factors for the TB performance, the studies extended existing research methods by measuring children's recursive ToM and syntactic recursive abilities. Results revealed that children's syntactic recursion abilities and their recursive ToM understanding (Study 1 and 3) and recursive ToM production (Study 2) correlated substantially–indicating that they tap a common underlying capacity for recursive thinking. However, the results regarding the main research question remain inconclusive: Study 1 suggests that children's recursive ToM, but not their advanced pragmatics understanding or general recursive thinking abilities, predict their TB performance. This relationship, though, could not be replicated in Studies 2 and 3 in which neither recursive ToM nor recursive thinking in general explained children's performance in the TB task.

The studies overall, therefore, do not provide clear evidence for the pragmatic analyses of children's performance recovery in the TB task around age eight to ten. Of course, absence of evidence for a solid relationship between advanced pragmatics understanding, recursive ToM and recursion in general and TB performance does not amount to evidence of absence of any such relationship. There might be an actual relationship between children's advanced pragmatics and related factors and their TB performance that could not be reliably shown in the current set of studies because of different reasons. There are different possibilities regarding how this may be the case about which we can here only speculate.

One possibility is that there is an actual relationship, at least, between recursive ToM understanding and TB performance as shown in Study 1, that is indeed less pronounced than Study 1 indicated. The sample size calculations for the subsequent studies were based on the potentially overestimated medium effect size from Study 1 which possibly caused Studies 2 and 3 to be too underpowered to detect an effect that may be more subtle. Future studies need to address this potential issue with adequate sample sizes that would also detect smaller relationships.

Another possibility is that we failed to detect an actual relationship because of the implementation of the various predictors, especially children's advanced pragmatic understanding. This was operationalized as children's understanding for indirect speech acts, more specifically, their understanding of metaphors and irony–based on approaches that suppose non-literal language comprehension involves the ascription of complex intentions. The pragmatic analysis of the TB tasks predicts that, by ascribing such complex communicative intentions, children at the end of the U-shape curve resolve their pragmatic confusions. However, much daily non-literal language including frozen metaphors (whose metaphorical content is "dead") and frequently used ironic remarks is conventionalized in language [24, 41]. In cases of such conventionalized metaphor use as "The ATM swallowed my credit card" the hearer usually understands what is said without processing the literal meaning first and making inferences about the speaker meaning in a second step [42 p. 116]. Similarly, ironic utterances that are used with high frequency and familiarity (e.g., "That's just great") might become, to some degree, conventionalized. As a consequence, their non-literal meaning might become directly accessible [41], (*see also* [43] for conventionalized versus non-conventionalized indirect requests). In contrast, understanding un-conventionalized non-literal language requires the hearer to make pragmatic inferences based on her world knowledge, the context and the lexical meaning of the utterance [24 p. 247] and therefore tap a different set of cognitive skills than conventionalized non-literal language does [44].

Another possibility is that we failed to detect a relation to children's non-literal language understanding since the relevant pragmatic abilities involved in the comprehension of the TB task, on the one hand, and of irony and metaphor comprehension, on the other hand, might be quite different. (We thank the anonymous Reviewer of PloS ONE for sharing this issue in their review of an earlier version of this manuscript). Ironic and metaphoric speech acts involve a mismatch of sentence and speaker meaning. The pragmatic challenge is that the

hearer needs to overcome the literal sentence meaning to be able to grasp the speaker meaning. In the TB task, however, sentence and speaker meaning are aligned. The challenge of the TB task might rather be to understand the speaker's conversational goal as the task and the answer to the test question are obvious and common ground and do not allow any other perspective on the scenario. Both the understanding for non-literal language and the TB task might be applications of advanced pragmatics understanding. However, they might not necessarily be seen as a unified phenomenon nor be subject to a same development [45].

Future studies, therefore, need to carefully operationalize these predictors of advanced pragmatics in order to measure children's abilities to ascribe recursive communicative intentions in valid ways. To this end, future work will need to test whether an operationalization with non-conventionalized non-literal language (such as novel metaphors and unfamiliar ironic utterances) or other pragmatics measures (e.g., understanding academic test question in general, understanding of literal meaning that changes with variations in context etc.) may succeed in measuring children's abilities for pragmatic inferences and recursive intention ascription and finding a relationship to children's TB performance.

Further future studies might test children's abilities in ascribing recursive intentions beyond the scope of (non-literal) verbal language understanding. An alternative that completely avoids any problems of interference of individual language knowledge would be to measure children's abilities in ascribing recursive communicative intentions in non-verbal scenarios. That might test the capacity for recursive, higher-order mindreading (that may also be applied to pragmatic language use) more generally. Cooperative coordination scenarios such as, for example, *Stag Hunt* scenarios, provide an elegant opportunity to ask children to reason recursively about other's non-verbal communicative acts. Stag Hunt scenarios [based on a parable by Jean-Jacque Rousseau, see 46] are game-theoretic interactions that require participants to cooperate with their fellow players to achieve a joint goal. Typically, they represent situations in which a player can choose between two options: the player can either hunt a *hare* (i.e. win a low value price) on her own or hunt a *stag* (i.e. win a high value price). If she decides to hunt a hare, the player will succeed independently of the other player's decision; if she decides to hunt a stag, in contrast, she only succeeds if the other player does so as well [46]. In a child-friendly adaptation, Wyman and colleagues showed that preschoolers succeed in coordinating with an adult co-player based on minimal non-verbal communication [47]. As the high-value option was only occasionally available, children needed to base their decision not only on what they themselves saw ("There is a stag") but also on what they thought about their fellow player ("She saw that there is a stag") and on what they inferred what their fellow player potentially thought about the them ("She knows that I know that she knows that there is a stag"). Children inferred the mutual knowledge of what their fellow player saw, knew and intended based on minimal non-verbal communicational cues (eye contact and smiling) and successfully based their decision for one of the two options on these inferences [47]. In a structurally related task by Grueneisen and colleagues, children were tested in a peer coordination scenario [48]. Children had to anticipate their partner's game decision based on a second-order mental state ascription ("She does not know that I know that *p*. She thinks that I falsely believe that *q*.") and had to coordinate their own game decision accordingly. Six-year-olds demonstrated their capacity to use such second-order mental state attributions to successfully coordinate with their peer without any communication [48]. Future studies could be based on such methods and thereby ask children to coordinate non-verbally in game scenarios with increasing complexity. Such adaptations might involve multiple co-players and, therefore, require even more complex recursive inferences (e.g., "Player 1 thinks that Player 2 did not see that Player 3 saw that there is a stag"). A systematic analysis of children's decisions in such game scenarios might then indicate the level of recursion at which they are able to reason.

A further line of future research will need to address the more general question of a unitary, domain-general development of recursive capacities. The current studies tested for children's recursive syntactic abilities until fourth order of recursion and children's recursive ToM understanding until fifth order of recursion as well as children's recursive ToM production in bluff scenarios. The results of correlated performance in recursion of mental representations and syntactic recursion provide first evidence for a shared underlying ability of general recursive thinking. In future research, these preliminary results need to be validated and extended over various forms of recursive thinking. Recently, it was theorized that embedding of recursive temporal representations shares conceptual similarities with embedding of recursive mental state representations [36]. Future studies thus need to compare children's development in holding recursive temporal representations (i.e. different forms of mental time travel) and higher-order ToM. Potential parallel trajectories of increasing levels of reasoning independently of its domain (i.e. mental states or temporal representations) would indicate shared underlying capacities of general recursive reasoning.

## Conclusion

In conclusion, the current set of studies replicate the typical TB performance pattern in children between six and ten years in an online testing format but do not provide clear evidence for an underlying development in advanced pragmatics that can explain this pattern. Future studies need to investigate more thoroughly whether this absence of evidence marks evidence of absence of any such relation between pragmatics and TB performance; or whether such relations exist but can be tapped only with suitably modified measures.

## Supporting information

**S1 File.**
(PDF)

**S1 Data.**
(XLSX)

**S2 Data.**
(XLSX)

**S3 Data.**
(XLSX)

## Acknowledgments

We thank Marlen Kaufmann for the organization of data collection, Lia Künnemann for the help with data collection and Cathrin Degen and Mellory Kripzak for help with reliability coding.

## Author Contributions

**Conceptualization:** Lydia Paulin Schidelko, Marina Proft, Hannes Rakoczy.

**Data curation:** Lydia Paulin Schidelko.

**Formal analysis:** Lydia Paulin Schidelko.

**Investigation:** Lydia Paulin Schidelko.

**Methodology:** Lydia Paulin Schidelko, Marina Proft, Hannes Rakoczy.

Project administration: Lydia Paulin Schidelko.

Resources: Hannes Rakoczy.

Supervision: Marina Proft, Hannes Rakoczy.

Visualization: Lydia Paulin Schidelko.

Writing – original draft: Lydia Paulin Schidelko.

Writing – review & editing: Marina Proft, Hannes Rakoczy.

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
