## [Decision Letter · Decision Letter 0]

16 Dec 2021

PONE-D-21-34148How do children overcome their pragmatic performance problems in the True Belief Task? The Role of Advanced Pragmatics and Higher-order Theory of MindPLOS ONE

Dear Dr. Schidelko,

Thank you for submitting your manuscript to PLOS ONE. I have sent it to two expert reviewers and have now received their comments back. As you can see at the bottom of this email and attached, the reviewers are quite positive about the manuscript. Both reviewers think that the paper addresses an interesting topic and that the studies are sound and informative. I do agree with their assessment. However, you will see that the reviewers also have some suggestions for improvement, notably in the introduction and discussion. I encourage you to take into account the reviewers' comments in a revised version of the manuscript.

We look forward to receiving your revised manuscript.

Kind regards,

Jérôme Prado

Academic Editor

PLOS ONE

Journal Requirements:

Reviewers' comments:

Reviewer #1: Summary

In this paper, Schidelko, Proft, and Rakozcy investigate possible mechanisms underlying the U-shaped curve in true belief task performance between the ages of 5 and 10. Earlier research suggests that initial failures in TB tasks is due to pragmatic difficulties when interpreting the test question. However, little is known about the developmental processes that lead older children to eventually overcome these difficulties and pass the task. In keeping with the pragmatic development explanation of children’s earlier failures, the authors aimed to test the hypothesis that children’s later successes on TB tasks are due to the development of more advanced pragmatic abilities. To test this hypothesis, the authors measured the relationship between TB performance and several different measures of general pragmatic competence. The authors ultimately find no reliable relationship between these measures of pragmatic ability and TB tasks. Possible explanations for these null results and future directions are discussed.

Overall evaluation

This was a well-designed study and a good attempt at explaining a very puzzling phenomenon within the theory of mind development literature. As someone who follows this literature closely, I found the manuscript informative, and I expect that sharing these null results will still ultimately contribute to our understanding of the phenomenon in question.

I do think that the introduction and discussion could use some work. While the empirical results of the study are inconclusive, there is an opportunity for a rich theoretical discussion, which I would encourage the authors to expand upon.

Comments

I found the section in the introduction on pragmatic development a little thin. At the very least this section should provide some kind of survey of how metaphor and irony-comprehension develop in neurotypical populations, since these abilities are being used as a proxy for advanced pragmatic knowledge (e.g. Lecce, S., Ronchi, L., Del Sette, P., Bischetti, L., & Bambini, V. (2019). Interpreting physical and mental metaphors: Is Theory of Mind associated with pragmatics in middle childhood?. Journal of Child Language, 46(2), 393-407). This will better motivate the choice of measures and also help the readers contextualize the results in the broader literature. Relying on a lone citation to Happe’s 1993 paper on metaphor comprehension in ASD is also problematic, as there is more recent evidence that certain forms of pragmatic processing are actually intact in ASD (see Geurts, B., Kissine, M., & van Tiel, B. (2019). Pragmatic reasoning in autism. In Thinking, reasoning, and decision making in Autism (pp. 113-134). Routledge.)

In the initial discussion of recursive theory of mind, it would also be helpful to explain how or why higher-order recursive mindreading would be related to TB task performance specifically, rather than just positing it as another general measure of pragmatic competence. Otherwise the choice of task feels slightly undermotivated. This might be accomplished by explaining how recursive mindreading is implicated in TB test-question responses.

In the discussion, the authors discuss some of the limitations of the metaphor-comprehension task as a measure of pragmatic understanding. One such limitation is the possibility that metaphorical expressions become highly conventionalized, so that their literal meaning is no longer semantically accessible. This is plausible, as far as it goes. However, this explanation does not extend to the irony comprehension measure of advanced pragmatic ability, which also failed to correlate with performance on the TB task. The meaning of ironic speech acts seems much more context-dependent than the meaning of metaphors, and much less amenable to conventionalization. So we are still left without much of an explanation of this particular result.

My two cents on the matter is that the relevant pragmatic abilities involved in understanding academic questions on the one hand and irony and metaphor on the other seem quite different. What makes irony (and perhaps some metaphors) hard is that the primary intention of these speech acts – i.e. the speaker meaning – doesn’t correspond with their secondary intention or sentence meaning. But the speaker meaning and sentence meaning in the TB test question are aligned. The issue is instead related to the fact that the answer to the question is already in the common ground, and so it’s not obvious to the child what the speaker’s conversational goal is. While there is a broad sense in which both of these obstacles are pragmatic in nature, it’s not obvious that what you need to know in order to overcome these obstacles is the same in each case, or that they would stem from similar kinds of experiences. Tentatively, one might even conclude that using metaphor and irony comprehension as a general measure of advanced pragmatic abilities might be misguided, or that pragmatic ability is not a unified phenomenon (see again the Geurts et al. chapter cited above).

The other suggestion in the discussion that future studies might focus on purely nonverbal forms of communication also struck me as odd given that children’s initial failures really do seem to be so closely tied to their understanding of the experimenter’s speech act. There are no trivial responses or academic questions in a nonverbal stag hunt game, and it’s not terribly obvious what the one thing has to do with the other. This is not to say that looking into the relationship between these tasks and TB tasks would be fruitless (perhaps it is a better measure of recursive mindreading), but it seems a bit odd for this to occupy so much space in the discussion.

On a related note, I found myself wondering whether this failure to detect a reliable correlation with more distally related pragmatic abilities might suggest a different future direction: finding a way to operationalize the specific type of pragmatic competency that underlies TB task performance, according to the pragmatic development account (e.g. understanding of trivial or academic questions). This would be in line with the broader theory

Minor correction:

On p. 9, a contrast is drawn between children’s recursive mindreading abilities as measured in Liddle and Nettle and adults’ recursive mindreading abilities as measured by O’Grady et al. It is implied that between 10-11 years of age and adulthood, we go from chance on fourth-order recursive ToM tasks to success on 7th-order recursive ToM tasks. But O’Grady and colleagues use a very different, “implicit” measure of recursive mindreading, so these two results are not directly comparable. This should be corrected.

Reviewer #2: The comments to the Author are included in the attached file labelled "Review". Please refer to this file. This provides my suggestions to revise the manuscript before publication. My overall assessment is " Minor revisions".

6. PLOS authors have the option to publish the peer review history of their article (what does this mean?). If published, this will include your full peer review and any attached files.

Reviewer #1: No

Reviewer #2: **Yes: **Diana Mazzarella

---

## [Author Response · Author response to Decision Letter 0]

28 Jan 2022

Reviewer #1

Summary

In this paper, Schidelko, Proft, and Rakozcy investigate possible mechanisms underlying the U-shaped curve in true belief task performance between the ages of 5 and 10. Earlier research suggests that initial failures in TB tasks is due to pragmatic difficulties when interpreting the test question. However, little is known about the developmental processes that lead older children to eventually overcome these difficulties and pass the task. In keeping with the pragmatic development explanation of children’s earlier failures, the authors aimed to test the hypothesis that children’s later successes on TB tasks are due to the development of more advanced pragmatic abilities. To test this hypothesis, the authors measured the relationship between TB performance and several different measures of general pragmatic competence. The authors ultimately find no reliable relationship between these measures of pragmatic ability and TB tasks. Possible explanations for these null results and future directions are discussed.

Overall evaluation

This was a well-designed study and a good attempt at explaining a very puzzling phenomenon within the theory of mind development literature. As someone who follows this literature closely, I found the manuscript informative, and I expect that sharing these null results will still ultimately contribute to our understanding of the phenomenon in question.

I do think that the introduction and discussion could use some work. While the empirical results of the study are inconclusive, there is an opportunity for a rich theoretical discussion, which I would encourage the authors to expand upon.

Comments

1. I found the section in the introduction on pragmatic development a little thin. At the very least this section should provide some kind of survey of how metaphor and irony-comprehension develop in neurotypical populations, since these abilities are being used as a proxy for advanced pragmatic knowledge (e.g. Lecce, S., Ronchi, L., Del Sette, P., Bischetti, L., & Bambini, V. (2019). Interpreting physical and mental metaphors: Is Theory of Mind associated with pragmatics in middle childhood?. Journal of Child Language, 46(2), 393-407). This will better motivate the choice of measures and also help the readers contextualize the results in the broader literature. Relying on a lone citation to Happe’s 1993 paper on metaphor comprehension in ASD is also problematic, as there is more recent evidence that certain forms of pragmatic processing are actually intact in ASD (see Geurts, B., Kissine, M., & van Tiel, B. (2019). Pragmatic reasoning in autism. In Thinking, reasoning, and decision making in Autism (pp. 113-134). Routledge.)

Answer to Comment 1: Thank you for this comment which is in line with the comments provided by Reviewer 2. In response, we added a more explicit description of the pragmatic analysis of children’s performance in the TB task and its parallels to the understanding of non-literal language; and we inserted a new short overview of the available experimental evidence in developmental non-literal language comprehension in the more detailed part “Advanced Pragmatic Understanding” in the Introduction. In the General Discussion, we partly revise and extend this analysis presented in the introduction as it became more fine-grained over the course of the studies. Thank you very much for providing us with several references to improve our manuscript.

2. In the initial discussion of recursive theory of mind, it would also be helpful to explain how or why higher-order recursive mindreading would be related to TB task performance specifically, rather than just positing it as another general measure of pragmatic competence. Otherwise the choice of task feels slightly undermotivated. This might be accomplished by explaining how recursive mindreading is implicated in TB test-question responses.

Answer to Comment 2: We now elaborate that children’s recursive mindreading abilities might be related to their TB performance in two ways (see page 11). On the one hand, recursive mindreading enables children to understand initially pragmatic speech acts (here the academic test questions) as they are able to ascribe complex communicative intentions. On the other hand, children might be still confused by the TB test question but use their recursive mindreading abilities to resolve this confusion by ascribing higher-order mental states (e.g., to the experimenter). We now discuss both possibilities in the manuscript and hope that the motivation for the thorough choice of tasks becomes more evident now. Thank you very much for helping us to improve our line of argumentation. 

3. In the discussion, the authors discuss some of the limitations of the metaphor-comprehension task as a measure of pragmatic understanding. One such limitation is the possibility that metaphorical expressions become highly conventionalized, so that their literal meaning is no longer semantically accessible. This is plausible, as far as it goes. However, this explanation does not extend to the irony comprehension measure of advanced pragmatic ability, which also failed to correlate with performance on the TB task. The meaning of ironic speech acts seems much more context-dependent than the meaning of metaphors, and much less amenable to conventionalization. So we are still left without much of an explanation of this particular result.

Answer to Comment 3: Thank you very much for pointing out that the explanation was not clear enough regarding children’s irony comprehension which also failed to correlate with the TB performance. We now elaborate that ironic utterances might also become conventional when used with high frequency and familiarity (e.g., “That’s just great!” or in German language “Du bist ja clever” meaning “You are [German modal particle] clever”, which is similarly used as ironic utterance in one of the two story lines in Exp. 1). Whether and which ironic utterances are conventionalized, might differ between children. So, this is only one speculative explanation for the missing relationship of TB performance and metaphor and irony comprehension. Future work will therefore need to continue to investigate, first, whether the development of comprehension of familiar and unfamiliar ironic utterances does differ (see Burnett, 2015 for differences in comprehension of speaker meaning but not of speaker attitude). Second, and in direct relation to our project, future studies will need to test whether an operationalization with rather unfamiliar ironic utterances and novel metaphors may succeed in measuring children’s abilities for pragmatic inferences and recursive intention ascription and finding a relationship to children’s TB performance. This is stated with reference to both irony and metaphor understanding in the new version of our manuscript (see page 45-46).

4. My two cents on the matter is that the relevant pragmatic abilities involved in understanding academic questions on the one hand and irony and metaphor on the other seem quite different. What makes irony (and perhaps some metaphors) hard is that the primary intention of these speech acts – i.e. the speaker meaning – doesn’t correspond with their secondary intention or sentence meaning. But the speaker meaning and sentence meaning in the TB test question are aligned. The issue is instead related to the fact that the answer to the question is already in the common ground, and so it’s not obvious to the child what the speaker’s conversational goal is. While there is a broad sense in which both of these obstacles are pragmatic in nature, it’s not obvious that what you need to know in order to overcome these obstacles is the same in each case, or that they would stem from similar kinds of experiences. Tentatively, one might even conclude that using metaphor and irony comprehension as a general measure of advanced pragmatic abilities might be misguided, or that pragmatic ability is not a unified phenomenon (see again the Geurts et al. chapter cited above).

Answer to Comment 4: We follow your thoughts on the distinction of pragmatic abilities that are involved in irony (and metaphor) understanding versus pragmatic abilities that are involved in the understanding of the speech act in the TB test. We recently had similar concerns; however, we did not concentrate too much on this difference before we conducted the study as the field of pragmatics comprises a wide spectrum and allows for various foci of analyses. Another level of analysis is, for example, to compare illocutionary and perlocutionary aspects of the TB test question and metaphoric/ironic utterances. However, as we cannot include all of these analyses in the discussion of our manuscript, we now decided to add the issue you raised as a further discussion point in the manuscript and emphasize the need of future work on this (see page 46). We want to thank you for sharing your thoughts on that with us so that we were able to elaborate this in our manuscript.

5. The other suggestion in the discussion that future studies might focus on purely nonverbal forms of communication also struck me as odd given that children’s initial failures really do seem to be so closely tied to their understanding of the experimenter’s speech act. There are no trivial responses or academic questions in a nonverbal stag hunt game, and it’s not terribly obvious what the one thing has to do with the other. This is not to say that looking into the relationship between these tasks and TB tasks would be fruitless (perhaps it is a better measure of recursive mindreading), but it seems a bit odd for this to occupy so much space in the discussion.

On a related note, I found myself wondering whether this failure to detect a reliable correlation with more distally related pragmatic abilities might suggest a different future direction: finding a way to operationalize the specific type of pragmatic competency that underlies TB task performance, according to the pragmatic development account (e.g. understanding of trivial or academic questions). This would be in line with the broader theory

Answer to Comment 5: Thank you very much for raising this crucial point. It made us realize that we did not introduce the relevant background and rational in sufficiently clear ways in the previous version. It was indeed misleading to present non-verbal measurements as an alternative measure for children’s verbal pragmatic language understanding. What we actually wanted to argue (and what we argue for now in much clearer ways) was the following hierarchical structure: non-literal language understanding might be one particular instance of recursive communitive intention ascription more generally, and that another option to test this more general capacity of recursive intention ascription that avoid inferences of individual language knowledge might be non-verbal Stag Hunt Scenarios. We now changed this in the manuscript and make clear that this is an alternative measure of recursive, higher-order mindreading (the same capacity that might also be applied in non-literal language understanding) and not an alternative measure of verbal non-literal language understanding itself (see page 46-47). Thank you very much for helping us to make this clearer for the reader. We hope that – taking these considerations and improvements into account – it is suitable to occupy some space for this in the discussion.

Minor correction:

6. On p. 9, a contrast is drawn between children’s recursive mindreading abilities as measured in Liddle and Nettle and adults’ recursive mindreading abilities as measured by O’Grady et al. It is implied that between 10-11 years of age and adulthood, we go from chance on fourth-order recursive ToM tasks to success on 7th-order recursive ToM tasks. But O’Grady and colleagues use a very different, “implicit” measure of recursive mindreading, so these two results are not directly comparable. This should be corrected.

Answer to Minor Comment/ Comment 6: Thank you for pointing out that this indeed needs to be reported in more nuanced ways. O’Grady and colleagues do actually use both explicit and implicit presentation of material (video clips of social interactions versus narrated story lines) and both explicit and implicit answer formats (video clips of social interactions versus read out recursive answer sentences). They conclude that adults are able to mindread to at least seven levels of embedding, both explicitly and implicitly. However, their data suggest that mindreading may be easier when stimuli are presented implicitly rather than explicitly (O’Grady et al., 2015). We added this information that O’Grady et al. (2015) in comparison to earlier reported studies (e.g., Liddle & Nettle, 2006) used both implicit and explicit stimuli and that children performed higher when stimuli were presented implicitly (see page 12).

Reviewer #2

Summary

The manuscript presents three experimental studies with 6- to 10-year-olds. The aim of these experiments is to assess whether children’s performance in a standard True-belief (TB) task can be predicted by their pragmatic skills, their recursive ToM skills, or more general recursive abilities. 

This investigation addresses an open question in developmental research: why do children’s performance in the TB task is characterized by a U-shaped developmental trajectory? More specifically, it focuses on the question of which cognitive skills determine the end of the U- shaped performance curve. 

Overall evaluation

The studies are methodologically rigorous and well described, the statistical analyses are appropriately reported, and they certainly deserve to be published in PLOS ONE. 

I have some recommendations to revise the manuscript before publication, which I list below. These concern the Introduction and the General discussion, and more specifically, the presentation and discussion of the notion of ‘advanced pragmatic abilities’, and their alleged role in the experimental puzzle under discussion. 

Comments

1. First of all, I believe the introduction should include a more explicit description of the pragmatic analysis of children’s performance in the TB task, and the extent to which the analysis provided meshes well with the available experimental evidence in developmental pragmatics. The authors suggest that “once children develop the prerequisite ToM capacities, they develop an understanding for pragmatics” (149-150) Before that, they would “mostly use and interpret language literally” (139). This analysis offers an overly-simplified picture of pragmatic development. On the one hand, recent work in pragmatics shows that children can interpret language non-literally before the age at which they typically pass the standard FB task. For instance, work from Nausicaa Pouscoulous and colleagues has shown that children are able to understand metaphors already at the age of 3 (see, e.g., Pouscoulous & Tomasello, 2020). Similar results have been found when looking at metonymy understanding, which also displays an interesting U-shaped curve (see, e.g., Falkum, Recasens & Clark, 2017; Köder & Falkum, 2020). On the other hand, some pragmatic phenomena, such as irony understanding, emerge later in development (for a discussion, see Pexman, forthcoming). Given the relevance of pragmatic skills to the issue under discussion, I suggest reviewing in more detail the experimental literature on the development of pragmatics. Most importantly, the authors should give more prominence to the distinct developmental trajectories that characterize different pragmatic phenomena, without assimilating metaphor and irony. This issue is particularly salient in the section “Advanced pragmatics understanding” where these phenomena are discussed under the umbrella term “indirect speech acts”, which would “involve a mismatch of sentence and speaker meaning” (208-209). 

Answer to Comment 1: The picture of the pragmatic analysis presented in these lines is indeed simplified in order to present the general idea in a nutshell. However, we now explicitly point to this simplification and also point to evidence that contradicts the simplified version of the idea. Furthermore, we added a more explicit description of the pragmatic analysis of children’s performance in the TB task and its parallels to the understanding of non-literal language as well as a short overview of the available experimental evidence in developmental non-literal language comprehension in the more detailed part “Advanced Pragmatic Understanding” in the Introduction. In the General Discussion, we partly revise and extend this analysis as it became more fine-grained over the course of the studies. We appreciate that you provided us with several sources to improve our manuscript. 

2. Furthermore, this section includes a very brief mention of the literature in clinical pragmatics and suggests that the pragmatic difficulties shown by individuals with ASD are fundamentally related to their ToM impairments. As this issue is vigorously debated in the current pragmatic literature, I suggest a more nuanced and informed discussion of the relevance of data from ASD to the issue under discussion (see, e.g., Andrés-Roqueta & Katsos, 2017; Kissine 2021; Mazzarella & Noveck, 2021). 

Answer to Comment 2: Thank you for this comment. We tried to present a more nuanced and up-to-date discussion of the current literature on neurotypical development of pragmatics in general (see answer to comment 1). With regard to pragmatics in clinical populations specifically, we tried in a first step to present an extensive overview of empirical findings as well as a summary of the ongoing debate in the current pragmatic literature. However, it then became apparent that this made the paragraph about pragmatics in clinical population very long in relation to the rest of the part about Advanced Pragmatic Understanding as the relationship between impairments in Theory of Mind and pragmatic understanding in clinical populations is not as conclusive as we presented it to be in the earlier version of our manuscript. We therefore now changed the manuscript in that we focus on the theoretical considerations and empirical findings on pragmatics development in neurotpyical populations only as it has yet to be clarified whether the relation between Theory of Mind abilities and pragmatics in neurotypical and neuroatypical development is comparable at all. (see page 10). 

3. Finally, I think the authors should more convincingly discuss what kind of advanced pragmatic abilities are expected to be involved in the TB task and why, and, in light of this, better justify the choice of the selected measurements (metaphor and irony understanding).

Answer to Comment 3: Thank you very much for this comment. Reviewer 1 mentions a similar point in their review (see comment 4 by Reviewer 1). We now review what kind of pragmatic abilities might be involved in the TB task and how these abilities might diverge from the pragmatic abilities involved in metaphor and irony understanding (see page 46). We also discuss in the new version of our manuscript that the development of these different underlying abilities might diverge as advanced pragmatic development cannot be necessarily seen as a unified phenomenon (see e.g., Geurts et al., 2019). It is thus subject to future work to examine both theoretically and empirically which kind of advanced pragmatic abilities are involved in the TB task. We therefore now consider in our new version of the manuscript a broad range of future studies including measures of non-literal speech acts as well as other measures of verbal advanced pragmatic understanding and of non-verbal communication. 

4. These changes in the Introduction should also be reflected in a revision of the General discussion. 

Answer to Comment 4: We addressed the issues discussed above also in the General Discussion. 

Minor comments:

5. 126-129: It would be useful to include more references to empirical data showing that “academic question formats are difficult to grasp for young children”. 

Answer to Minor Comment/ Comment 5: Thank you. Unfortunately, the main empirical source usally cited in this context is an unpublished manuscript (Pemer, J., Leekam, S. R., & Wimmer, H. (1986). The insincerity of conservation questions: Children’s growing insensitivity to experimenters’ epistemic intentions). We therefore included some other references that show that the epistemic status of the interviewer can have an impact on the children’s answering behavior (see page 5). 

6. Fig. 2 The visual animation seems very difficult to interpret, and it does not strike as an intuitive representation of the story “the video dilemma”. Could you please better justify the choice of this visual representation format and spell out more explicitly its appropriateness for the target age groups 

Answer to Minor Comment/ Comment 6: Thank you for this comment. We changed Figure 2 in order to show more clearly how the (indeed very long and complex) answer options were presented appropriately for the target age group (see page 19). Children heard the sentences presented slowly and sectional (in parts) while the according protagonists and their thoughts were represented visually on screen. Children were allowed to hear the answer options up to four times. Figure 2 represents a rather complex example (Level 4 mental state reasoning). Only older children were expected to solve this question. Before that point during the test session, children already heard the second- and third-order mental state questions as well as control questions in which they got used to the format as they heard stepwise longer and longer sentences each time. 

Typos

• 354: Delete “were” - done

• 717: Delete “that” - done

---

## [Decision Letter · Decision Letter 1]

31 Mar 2022

How do Children Overcome Their Pragmatic Performance Problems in the True Belief Task? The Role of Advanced Pragmatics and Higher-order Theory of Mind

PONE-D-21-34148R1

Dear Dr. Schidelko,

We’re pleased to inform you that your manuscript has been judged scientifically suitable for publication and will be formally accepted for publication once it meets all outstanding technical requirements.

Kind regards,

Jérôme Prado

Academic Editor

PLOS ONE

Additional Editor Comments (optional):

Reviewers' comments:

Reviewer's Responses to Questions

**Comments to the Author**

1. If the authors have adequately addressed your comments raised in a previous round of review and you feel that this manuscript is now acceptable for publication, you may indicate that here to bypass the “Comments to the Author” section, enter your conflict of interest statement in the “Confidential to Editor” section, and submit your "Accept" recommendation.

Reviewer #1: All comments have been addressed

2. Is the manuscript technically sound, and do the data support the conclusions?

Reviewer #1: (No Response)

3. Has the statistical analysis been performed appropriately and rigorously? 

Reviewer #1: (No Response)

4. Have the authors made all data underlying the findings in their manuscript fully available?

Reviewer #1: (No Response)

5. Is the manuscript presented in an intelligible fashion and written in standard English?

Reviewer #1: (No Response)

6. Review Comments to the Author

Reviewer #1: (No Response)

7. PLOS authors have the option to publish the peer review history of their article (what does this mean?). If published, this will include your full peer review and any attached files.

Reviewer #1: **Yes: **Evan Westra

---

## [Editor Report · Acceptance letter]

5 Apr 2022

PONE-D-21-34148R1 

How do Children Overcome Their Pragmatic Performance Problems in the True Belief Task? The Role of Advanced Pragmatics and Higher-order Theory of Mind 

Dear Dr. Schidelko:

I'm pleased to inform you that your manuscript has been deemed suitable for publication in PLOS ONE. Congratulations! Your manuscript is now with our production department. 

Kind regards, 

on behalf of

Dr. Jérôme Prado 

Academic Editor

PLOS ONE